# Effects of Self-Mastery on Adolescent and Parental Mental Health through the Mediation of Coping Ability Applying Dyadic Analysis

**DOI:** 10.3390/bs10120182

**Published:** 2020-11-27

**Authors:** Chiara Filipponi, Peter J. Schulz, Serena Petrocchi

**Affiliations:** Faculty of Communication, Culture and Society, Institute of Communication and Health, Università della Svizzera italiana, Lab Building, Via Buffi 13, 6900 Lugano, Switzerland; chiara.filipponi@usi.ch (C.F.); peter.schulz@usi.ch (P.J.S.)

**Keywords:** self-mastery, coping ability, mental health, adolescence, parent, actor-partner interdependence model extended mediation

## Abstract

Evidence demonstrated that self-mastery and coping ability predict mental health in adults and children. However, there is a lack of research analyzing the relationships between those constructs in parents and children. Self-report data from 89 dyads (adolescents’ mean of age = 14.47, SD = 0.50; parents’ mean of age = 47.24, SD = 4.54) who participated in waves 17, 18, and 19 (following T1, T2, and T3) of a nineteen-wave longitudinal study were analyzed using the Actor-Partner Interdependence Model’s extended Mediation. Results showed significant actor effects of parents’ and adolescents’ self-mastery (T1) on mental health (T3) and the mediator effect of their coping abilities in managing stress (T2). Both a higher parental education level and being a mother positively influenced adolescents’ coping ability. The mutually beneficial relationships between parents’ and adolescents’ self-mastery, coping ability, and mental health were not demonstrated. Self-mastery is a significant predictor of adolescents’ and parents’ mental health, and coping ability serves as a good mediator between them. Qualitative research may clarify reasons why partner effects in the model were found to be non-significant. Further research should re-test this model with a larger sample size during childhood, when parents provide significant behavioral models for their children—as well as in adolescence, considering the peer group—to develop guidelines for behavioral interventions.

## 1. Introduction

Maintaining good mental health is essential for assuring an individual’s well-being during the entire course of life. Individuals learn self-regulation abilities, such as self-efficacy and locus of control, during childhood through parental modeling [1]. Significant others (i.e., individuals who are deeply influential in one’s life, like friends or family members) also play an essential role in shaping self-perception and self-regulatory processes [2]. Self-efficacy, locus of control, and self-perception are incorporated in the concept of self-mastery [3], which is defined as the feeling of being able to overcome life challenges with personal effort [4].

Longitudinal evidence demonstrated that self-mastery measured during childhood is a valid predictor of mental health during adulthood [5]. Moreover, Infurna and Mayer [6], in a longitudinal research, confirmed that self-mastery in adults leads to good physical and mental health. Another study found that adolescents’ self-mastery, together with positive relations with significant others, helps them become resilient and cope with life challenges [3] more effectively. During adulthood, it was also found that self-mastery is associated with the ability to cope with stressful situations [7,8] even considering daily life stressors [9]. It is well known that stressful experiences, especially when repeated over time, are detrimental to human health [10,11]. Coping ability is a protective factor for mental health both during adulthood [12] and adolescence [13].

One gap in the literature is the lack of consideration of parents and their children’s mutually beneficial relationships. If self-mastery develops through parental modeling [1,14], research should also measure parental influence on children. Moreover, since children have an active role in the relationship with their parents, especially during adolescence, it should be expected that the relationship would be mutually beneficial or consequential (i.e., parent influences child, child influences parent). According to the Interdependence Theory, outcomes from dyadic relationships depend on the two individuals’ characteristics [15]. For example, research [16,17] has shown an interdependence between parents’ and adolescents’ attachment styles which establishes a conjoint effect on their perceived quality of relation.

The concept of self-mastery, however, even if it is a dyadic construct (i.e., an ability learned within relationships with significant others—e.g., parents), is not yet examined in a dyadic way. In the literature, researchers examined those constructs through analyses implying the independence of the data, such as the longitudinal mediation model [6] or Structured Equation Models (SEMs) [5]. We hypothesized that one individual’s self-mastery would be connected to the other’s mental health, either directly or through the coping ability mediation. These relations were predicted because both self-mastery and coping ability in managing stress serve as protective factors for good mental health [12,13]. Additionally, they positively influence one’s ability to handle one’s own stressful situations and to provide support to others through their stressful circumstances. Similarly, good mental health may be improved by an interdependence between a parent and a child. Adolescents with less stressed and well-adapted parents may experience a more positive family environment to mature and learn coping mechanisms for stressful situations. Such parents may also create more positive interactions with their children, provide improved familial relations, reduce levels of conflict, encourage shared decision-making, and maintain reciprocal trust. The aforementioned influences not only the children’s mental health but also parental mental health.

Therefore, the present study aimed to analyze the longitudinal mutually beneficial influences between parents’ and adolescents’ self-mastery, ability to cope with stress, and mental health through the application of the Actor-Partner Interdependence Model’s extended Mediation analysis (APIMeM) [18]. With the APIMeM, four actor effects (two direct and two indirect) and four partner effects (two direct and two indirect) can be estimated. Actor effects estimate whether the X variable of a person influences his or her score on the Y variable directly or through the mediation of a mediator M. Partner effects estimate whether the X variable of a person influences the score of the partner’s Y variable directly or through the mediation of a mediator M. Partner effect allows for the exploration of the interdependency across the individuals of a dyad. Figure 1 shows the theoretical model tested.

The present study addressed the following:

HP1a: Does a higher self-mastery ability significantly predict better mental health (i.e., direct actor effects), for both adolescents and adults?

HP1b: Does a higher coping ability serve as a mediator between self-mastery and mental health (i.e., indirect actor effects) for adolescents and adults?

RQ1: Are there mutual influences between parents’ and adolescents’ self-mastery, coping ability, and mental health over a three-year time period (i.e., direct and indirect partner effects)?

## 2. Methods

### 2.1. Participants and Procedure

Data came from the Swiss Household Panel (SHP) [19], a large annual panel study on a random sample of Swiss households. We included data of parent-adolescent dyads collected in 2015 (wave 17), 2016 (wave 18), and 2017 (wave 19; following T1, T2, and T3). The sample at T1 was composed of 256 dyads, with an attrition rate from T1 and T2 of 7% (T2: N = 239 dyads). At T3, the sample size was 259 dyads, with 17 dyads participating in the research at T1 and T3 and three new dyads only participating at T3. Data coming from the three new dyads added in T3 are missing in T1 and T2; therefore, we did not use their data for analyses. Data from the 17 dyads participating in T1 and T3 was removed because mediator scores were missing.

The initial sample was then composed of 239 dyads. Only dyads with missing data lower than 10% in all the main variables of interest were considered. For this reason, the final sample was composed of 89 dyads with full data in the three-time data collection. Mann–Whitney U tests comparing adolescents with completed data to those with uncompleted data showed no differences in self-mastery, coping ability, and mental health. Similarly, no differences were found between parents who completed the survey and those who did not in all the variables. For details, see Table 1.

In the final sample, the adolescents’ mean age was 14.47 (SD = 0.50; 48 males), and parents’ mean age was 47.24 (SD = 4.54, age-range 37–58; 59 = mothers; 30 = fathers). The mean of the highest education level for parents was 4.02 (SD = 1.04; range 2–6; (based on ISCED-classification) [20]. Eighty-two percent of parents were employed, 17% unemployed, and 1% retired. In regard to the marital status, 77% were married, and 23% were single parents (i.e., 18% divorced, 3% never married, and 2% widowed).

The data collection followed the ethical standards defined by the Declaration of Helsinki.

### 2.2. Measures

Self-mastery was measured at T1 with five items measuring self-perception [21,22,23] and six items measuring sense of control [23], as done before [3]. Levy and colleagues [21] adapted the first three items measuring self-perception (“I feel like I have little influence on the events of my life”; “I am easily overcome by unexpected problems”; and “In general, I have no difficulty choosing between two possibilities”) from Strodtbeck [23]. The other two items (“Sometimes I feel useless”; and “Finally, I am rather pleased with myself”) were adapted from the self-esteem scale by Rosenberg [22]. The six items measuring sense of control were as follows: “I can do just about anything I really set my mind to”; “When I really want to do something, I usually find a way to succeed at it”; “Whether or not I am able to get what I want is in my own hands”; “What happens to me in the future mostly depends on me”; “Other people determine most of what I can and cannot do”; and “I sometimes feel I am being pushed around”.

Response options ranged from 0 (“I completely disagree”) to 10 (“I completely agree”). The scores of the negatively worded items were reverse-coded. Two final scores were created—one for parents and one for adolescents—by calculating the mean of the items, with higher scores indicating greater self-mastery. For parents, the internal consistency was discrete (*α* = 0.80, *rs* > 0.24), whereas for adolescents it was acceptable (α = 0.68, *rs* > 0.26) after the exclusion of two items with inter-item correlation <0.20.

Coping ability to manage stress was measured at T2 with four items of the Perceived Stress Scale (PSS) [24]. Questions ask participants their feelings and thoughts perceived during the last month: “How often have you felt that you were unable to control the important things in your life?”; “How often have you felt confident about your ability to handle your personal problems?”; “How often have you felt that things were going your way (or going the right way for you)?”; and “How often have you felt difficulties were piling up so high that you could not overcome them?”. The items ranged from 1 (“never”) to 5 (“very often”). The scores of the negatively worded items were reverse-coded. Two final scores were created—one for parents (α = 0.62, *rs* > 0.33) and one for adolescents (α = 0.73, *rs* > 0.21, after the exclusion of one item)—by calculating the mean of the items. Higher scores indicate greater coping abilities.

Mental health was measured at T2 and T3 by six items. One item asked the frequency of negative moods experienced by the participants (“Do you often have negative feelings such as having the blues, being desperate, suffering from anxiety or depression?”), and another asked the frequency of positive attitudes characterizing participants (“Do you have plenty of strength, energy, and optimism?”). The other four items asked how frequently they generally experienced both positive emotions (joy) and negative emotions (anger, sadness, and worry). All the items ranged from 0 (“never”) to 10 (“always”). The scores of the negatively worded items were reverse-coded. Two final scores were calculated—one for parents and one for adolescents—by calculating the mean of the items, with higher scores indicating better mental health. The internal consistency of parents’ score was acceptable both at T2 (*α* = 0.77, *rs* > 0.34) and T3 (*α* = 0.81, *rs* > 0.45). After the exclusion of one item, the adolescents’ score of mental health showed acceptable internal consistency at T2 (*α* = 0.77, *rs* > 0.22) and T3 (*α* = 0.72, *rs* > 0.20).

### 2.3. Analysis Plan

Both hypotheses and the research question were analyzed on SPSS v.25 with the added module MEDYAD [25,26]. The goodness-of-fit indices were tested in Rstudio software v.1.2.5019, applying the Structural Equation Model (SEM) [18] with the Lavaan package [27].

## 3. Results

The correlation between adolescents’ and parents’ mental health was low-moderate (r = 0.22, *p* < 0.05). This correlation suggests a sufficient overlap between parents’ and adolescents’ scores on mental health and allows us to consider the dyad as the unit of analysis. Table 2 reports all the other correlations between the variables.

The APIMeM shows four significant actor effects with good fits of the model, where R^2^ = 0.65 for adolescents and R^2^ = 0.67 for parents. Adolescents’ high self-mastery increases mental health both directly (β = 0.45, *p* < 0.01) and indirectly (β = 0.69, *p* < 0.01) through the mediation of the coping ability to manage stress (β = 0.16, *p* < 0.05). Likewise, parents’ self-mastery significantly predicts their mental health both directly (β = 0.38, *p* < 0.01) and indirectly (β = 0.23, *p* < 0.01) through the mediation of the coping ability to manage stress (β = 0.94, *p* < 0.01). The paths were controlled for gender, mental health measured at T2, and parental educational level. As Figure 2 shows, the children of parents with a high level of education had a greater ability to manage stress. Additionally, being a mother positively influences the ability to manage stress. We did not find any significant partner effects; therefore, adolescents’ self-mastery did not have a positive effect directly or through the mediation of coping ability on their parents’ mental health. The same results were determined for parents. The model shows a good model fit: χ^2^ (11) = 6.14, *p* = 0.86, CFI = 1, RSMSEA = 0.000 (90-LOW = 0.000, 90-HIGH = 0.047), SRMR = 0.032. The estimated paths were as previously described and as reported in Figure 2.

## 4. Discussion

Our findings confirmed that both adolescents’ and parents’ self-mastery longitudinally promote their mental health, through a direct link and the mediation of coping ability to manage stress. We also found that self-mastery on mental health of one is not influenced by the other (i.e., parental self-mastery does not influence adolescents’ mental health, and vice versa). Likewise, adolescents’ self-mastery does not influence parental mental health. The same was found for coping ability in managing stressful situations. The relationship could be explained by the fact that, during childhood, parents are the primary significant figures on whom children model their behavior. However, during adolescence, significant influences derive from outside the family, especially from peers. By contrast, psychological traits, once developed, become stable over time [5], and if they become positive internal resources like self-mastery and coping ability, they then serve as protective factors for mental health [12,13] together with others [28,29].

We also found that higher parental education [30] and being a mother [31] positively influences adolescents’ coping ability in managing stress, as previously demonstrated. Further research should estimate the same model during childhood, a period of life in which psychological traits begin to develop as well as when parental influence is even more critical than in adolescence. Concerning adolescence, we suggest re-testing the same model considering the peer group, and other significant relationships for adolescents [32]. A deeper understanding of these factors can provide specific guidelines for intervention due to the role played by self-mastery and coping ability in assuring good mental health [5,6].

This study has some limitations. First, the measures were tested through self-report questionnaires. Participants’ answers could be biased by social desirability and metacognitive ability. Second, the sample size did not allow a full generalization of the results and thus biased the statistical power. Further research should consider a larger sample size of dyads and consider social desirability. Third, we could not control the autoregressive effects of adolescents’ self-mastery and coping skills due to the lack of measurements in two of three-time points. Forth, we found the alpha coefficients in the adolescents’ measures to be acceptable to moderate, compared to those of the parents. This aspect introduces a bias in the measures’ reliability and needs to be considered in the future. Finally, our study did not consider many possible confounders, such as socio-economic status and marital status besides gender, parental education, and age. Another point concerns the measurement of the coping ability to manage stress, that did not distinguish among different coping strategies.

Our research shows a relationship between self-mastery and mental health through the mediation of coping ability in both parents and adolescents. However, we did not confirm the hypothesized mutually beneficial relationships between parents’ and adolescents’ self-mastery, coping ability, and mental health across a three-year time period. A qualitative research may clarify the reasons why partner effects in the model were found as non-significant. One possible future application may consider re-testing our model with a larger sample size during childhood, when parents represent significant behavioral models for their children, and again in adolescence considering the peer group, to develop standardized guidelines for intervention.

## Figures and Tables

**Figure 1 behavsci-10-00182-f001:**
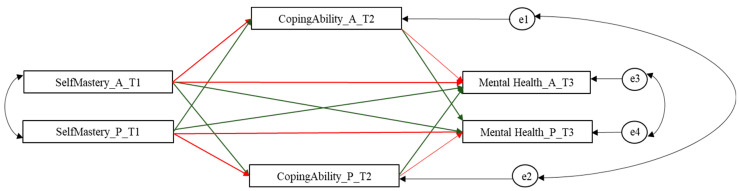
Theoretical model tested. Note: SelfMastery = Self-Mastery; CopingAbility = perceived coping ability to manage stress event-related; Mental_Health = Mental Health; A = adolescent; *p* = parent. Red lines indicate actor effects; green lines indicate partner effects.

**Figure 2 behavsci-10-00182-f002:**
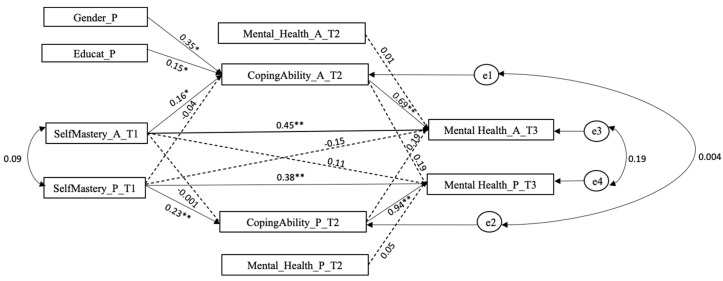
The estimated Actor-Partner Interdependence Model’s extended Mediation (APIMeM). Note: SelfMastery = Self-Mastery; CopingAbility = perceived coping ability to manage event-related stress; Mental_Health = Mental Health; A = adolescent; *p* = parent. Educat = parents’ level of education based on the ISCED classification [20]. Plain lines showed significant paths, dotted lines non-significant paths. * *p* < 0.05; ** *p* < 0.01. Source of the data: SHP.

**Table 1 behavsci-10-00182-t001:** Medians of the main variables and comparisons between continuers and non-continuers.

	Continuers (N = 89 dyads) Mdn	Non-Continuer (N = 150 dyads) Mdn	
*Parents*			
Self-mastery	7.27	7.91	U = 580.5, ns
Coping Ability	4.00	4.50	U = 371.5, ns
Mental Health	7.33	7.17	U = 878, ns
*Adolescents*			
Self-mastery	7.18	7.09	U = 305.5, ns
Coping Ability	4.00	4.00	U = 3025, ns
Mental Health	7.17	7.17	U = 6764, ns

Note. ns = non-significant. Source of the data: Swiss Household Panel (SHP).

**Table 2 behavsci-10-00182-t002:** Pearson’s Bivariate Correlations between all variables.

	n	M	SD	1	2	3	4	5	6
Adolescents’ Self-Mastery (1)	89	7.44	1.05	-					
Parents’ Self-Mastery (2)	89	7.37	1.12	0.09	-				
Adolescents’ Coping Ability (3)	89	4.00	0.68	0.22 *	−0.02	-			
Parents’ Coping Ability (4)	89	4.05	0.55	0.05	0.47 **	0.05	-		
Adolescents’ Mental Health (5)	89	6.89	1.30	0.47 **	−0.07	0.48 **	0.06	-	
Parents’ Mental Health (6)	89	7.14	1.30	0.19	0.51 **	0.12	0.57 **	0.22 *	-

Note: * *p* < 0.05; ** *p* < 0.01. Source of the data: SHP.

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
