# Peer review of "Effects of Self-Mastery on Adolescent and Parental Mental Health through the Mediation of Coping Ability Applying Dyadic Analysis"

_behavsci, 2020, doi:10.3390/bs10120182_

Round 1

Reviewer 1 Report

A structural quantification model analysis should be added with its statistics (c2, RMSEA, NCP, AIC, BIC, GFI, NFI, RMR, PNFI, PGFI, AGFI…).

You must attach documentation from an agency that approves the investigation since there is data collection with adolescents.

You must add references from the last three years (there is no appointment from 2018, 2019 or 2020)

Reviewer 2 Report

There are a few things that would improve this paper, which I think overall is quite well written, except for some problematic English expression issues which just need a good proof and edit.

I'm not sure where the assumption that parents and adolescents are partners comes from. The relationship is parent/child, and they are not equal and so why you would expect to see a mutual reinforcement of those influences is a mystery to me.

There are some limitations you havent considered and should.

1.First there most likely many more influences that the survey data doesnt capture.

2.Second the analysis doesn't explain why you found what you did--that would require more research, probably qualitative. I would doubt that you need to retest the model.

I think what you found is what most people would expect, and I think you have already pointed to literature that supports your findings. 

The paper's findings are of limited value, but may well be of use to researchers looking for confirmation of these findings. English needs some proofreading, but beyond my note on limitations, I don't see too many issues with this as a  short report.

Reviewer 3 Report

Overall this is a well conceptualized, methodologically sound, and appropriately analyzed  study. 

Additional information regarding the following will provide greater clarity for understanding  of the results and conclusions drawn by the authors.  

Introduction

Line 52-53,  the authors state that "the concept of self-mastery, even if it is a dyadic construct..."  Can the authors provide some clarity about what is meant by "even if it is a dyadic construct" ? Also, is it the author conclusion/assertion that the concept of self-mastery is not examined in a dyadic way. Is there cited literature to support their viewpoint stated here?  

Methods

Did the authors conduct the Swiss Household Panel (SHP) study?  If they did not, what procedures and approvals were they required to get in order to  access the data? and meet all research ethics guidelines/approval for use of the data? To make it clearer,  the authors should provide a statement indicating that that the authors' use of the data from that study also received the appropriate review and approval for use of the data for conducting analysis to answer the research questions posed in this study, particularly if they were not part of the research group collecting the original data. 

 The authors measured the key variable of self-mastery by combining items from various standardized instruments. The authors should provide rationale and/or citation(s) to support this practice and also clarify whether scaling (0 to 10) was the same across the various measures used or what adjustments were made with scaling in composite measure of self-mastery. 

Overall the authors should make it explicit that they developed the measures for this study from items available in the dataset. For instance, it is not totally clear whether the mental health measure developed by the authors (selecting the 6 items indicated) or was established measure included by those conducting the original Swiss Household Panel (SHP) study?

Analysis

The analysis and methods of analysis is appropriate for answering the main research questions.

Results/Discussion

Along with self-report as a limitation of the study as mentioned by the authors, could the authors discuss whether the way the authors combined items to measure the key variables may have been a factor in results observed as well as be considered as a limitation of the study as the resulting alphas were varied and no† particularly strong.

Round 2

Reviewer 1 Report

A structural quantification model analysis should be added with its statistics (Ji-Squared, RMSEA, NCP, AIC, BIC, GFI, NFI, RMR, PNFI, PGFI, AGFI…).

Author Response

Thank you for your suggestion. However, as discussed during the first round, the MEDYAD application gives results comparable to those obtained through SEM. Anyway, we have estimated the same model with RStudio, and now you can find a sentence with the goodness-of-fit indices for the model estimated.